# Easy-to-Operate Co-Flow Step Emulsification Device for High-Throughput Three-Dimensional Cell Culture

**DOI:** 10.3390/bios12050350

**Published:** 2022-05-18

**Authors:** Chunyang Wei, Chengzhuang Yu, Shanshan Li, Tiejun Li, Jiyu Meng, Junwei Li

**Affiliations:** 1Hebei Key Laboratory of Robotic Sensing and Human-Robot Interactions, School of Mechanical Engineering, Hebei University of Technology, Tianjin 300132, China; 201811201008@stu.hebut.edu.cn (C.W.); sli_mems@hebut.edu.cn (S.L.); 2State Key Laboratory of Reliability and Intelligence of Electrical Equipment, Hebei University of Technology, Tianjin 300130, China; 201811201004@stu.hebut.edu.cn (C.Y.); 201521202004@stu.hebut.edu.cn (J.M.); 3Jiangsu Key Laboratory of Advanced Food Manufacturing Equipment and Technology, Jiangnan University, Wuxi 214122, China; 4Institute of Biophysics, School of Health Sciences and Biomedical Engineering, Hebei University of Technology, Tianjin 300401, China; 5Department of Computer Science and Electrical Engineering, Hebei University of Technology, Langfang 065000, China

**Keywords:** microfluidics, droplet arrays, 3D cell culture, co-flow step emulsification, lab on a chip

## Abstract

Cell culture plays an essential role in tissue engineering and high-throughput drug screening. Compared with two-dimensional (2D) in vitro culture, three-dimensional (3D) in vitro culture can mimic cells in vivo more accurately, including complex cellular organizations, heterogeneity, and cell–extracellular matrix (ECM) interactions. This article presents a droplet-based microfluidic chip that integrates cell distribution, 3D in vitro cell culture, and in situ cell monitoring in a single device. Using the microfluidic “co-flow step emulsification” approach, we have successfully prepared close-packed droplet arrays with an ultra-high-volume fraction (72%) which can prevent cells from adhering to the chip surface so as to achieve a 3D cell culture and make scalable and high-throughput cell culture possible. The proposed device could produce droplets from 55.29 ± 1.52 to 95.64 ± 3.35 μm, enabling the diverse encapsulation of cells of different sizes and quantities. Furthermore, the cost for each microfluidic CFSE chip is approximately USD 3, making it a low-cost approach for 3D cell culture. The proposed device is successfully applied in the 3D culture of saccharomyces cerevisiae cells with an occurrence rate for proliferation of 80.34 ± 3.77%. With low-cost, easy-to-operate, high-throughput, and miniaturization characteristics, the proposed device meets the requirements for 3D in vitro cell culture and is expected to be applied in biological fields such as drug toxicology and pharmacokinetics.

## 1. Introduction

The standard two-dimensional (2D) monolayer culture grown on a Petri dish or in a plastic flask was widely employed for in vitro culture, while it cannot adequately replicate the characteristics of in vivo conditions. As a more advanced method, three-dimensional (3D) in vitro culture can accurately simulate in vivo environments and mimic cell–cell and cell–extracellular matrix (ECM) interactions [1,2,3]. In addition, experiments conducted according to the 3D culture model can provide more realistic predictions for safety and risk assessments [4,5].

Currently, various approaches have been proposed to achieve 3D cell cultures. The ultra-low attachment plate method [6] can regulate the space between the top plate and bottom substrate, while it cannot construct a 3D ECM microenvironment within a predefined space. Rotating bioreactors [7] provide a low-fluid shear growth environment, but additional rotating devices hinder the miniaturization and simplification of the system. Furthermore, the magnetic levitation method [8] can manipulate the geometry of a cell mass and realize the co-culture of multicellular clusterings; however, the magnetic iron oxide nanoparticles added in the ECM may hinder the cells’ proliferation and migration. 

In recent years, the emergence of microfluidics has provided an integrated and miniaturized platform for in droplet 3D cell culture with excellent manipulation [9], low reagent consumption [3], and integrated devices [10,11]. As shown in Figure 1a,b, the microfluidic hanging drop [12] method makes drops that hang on the bottom/side of a substrate so that cells can aggregate to the concave bottom of the droplets. However, it is difficult to achieve real-time observation of in drop cells; in addition, the reagent consumption levels of this system are still in the range of tens of microliters. More advanced microfluidic T-junction [13], flow-focusing [14,15], and coaxial flow [16,17,18] methods (Figure 1c–e) can produce small-sized droplets with low reagent consumption. However, the volume fractions of the resulting droplets are relatively small (<30%) [19], which results in low-density information in the microscope’s field and hinders scalable and high-throughput cell culture. Although the sparse droplets could be collected in an off-chip reservoir and allowed to “cream” due to their buoyancy [20], droplet-coalescence may occur due to dust, static charge, and flow through syringes, needles, and tubing.

In this paper, we present a novel microfluidic device for 3D cell culture based on “co-flow step emulsification” (CFSE) (Figure 1f) [21,22,23]. With the combination of “single-layer soft lithography” and “punching fabrication”, an elaborate “step” microstructure is easily constructed inside a microfluidic device, which is used to create a Laplace pressure gradient for breaking cell medium into discrete droplets [24]. In this CFSE device, when the co-flow (consisting of cell medium and oil) crosses the step, it pinches into emulsion droplets due to the Rayleigh–Plateau instability [25,26]. The resulting droplets have an ultra-high-volume fraction (72%), which makes for a more closely-packed droplet arrangement and benefits image recording [27,28]. Therefore, more droplet partitions can be optically observed within a limited field of view, thus achieving large-scale and high-throughput cell culture. In addition, compared to the commercial 3D culture plate, the proposed microfluidic device is economical and easy to operate—a benefit of the succinct “punching” chip fabrication method. Furthermore, the proposed cell culture method is flexible—a benefit of the regulable droplet diameters and scalable droplet numbers. Saccharomyces cerevisiae is the best-studied eukaryote and a valuable tool for most aspects of basic research on eukaryotic organisms; its unicellular nature makes it easily amenable to genetic manipulation in biological fields [29]. We successfully applied this system to saccharomyces cerevisiae cell culture, with a 200 min incubation time in monodisperse droplets (*Φ* = 50.15 ± 1.13 μm), the occurrence rate for the proliferation reached 80.34 ± 3.77%. With low-cost, easy-to-operate, scalable, and high-throughput characteristics, the proposed microfluidic device is expected to expand the range of 3D cell culture applications.

## 2. Materials and Methods

### 2.1. System Design

The proposed in droplet 3D cell culture system consists of three main parts, including a fluid control platform, a microfluidic CFSE chip, and an inverted microscope. The working mechanism of the proposed system is as follows: First, the cell suspension is prepared and stored in a container connected to the fluid control system. Second, the cell suspension is pumped into the microfluidic CFSE chip. By precisely controlling the flow rates of the two-phase fluid, co-flow step emulsification occurs; thus, the cell suspension is divided into monodisperse droplets for achieving cell encapsulation. Third, the in droplet cells are cultured in the chip. The culture process is videotaped or regularly photographed by an inverted microscope for in situ growth monitoring; the recorded data are used for subsequent analysis and evaluation.

### 2.2. Chip Fabrication

The microfluidic device comprises borosilicate glass and poly-dimethylsiloxane (PDMS) patterned with microchannels. Microfluidic master moulds of blue photosensitive dry film (ChangXing Chemistry, Tianjin, China) on 4-inch borosilicate glass wafers were fabricated in a clean-room facility using a microchannel mask (design with Auto CAD 2019 software, Autodesk, San Rafael, CA, USA). A PDMS replica was obtained by moulding PDMS (Sylgard-184, Dow-Corning, Midland, MI, USA) on the top of a dry film mould using standard single-layer soft lithography technology [30,31]. Oil and cell medium inlets were fabricated with a bio-puncher (OD = 1.6 mm). The step microstructure of the device was fabricated with a bio-puncher (OD = 4.0 mm). Lastly, the obtained PDMS replica was bonded with a borosilicate glass slide (25 × 75 × 1 mm) using oxygen plasma treatment. Before the cell culture experiment, the microfluidic device was autoclaved for sterilization.

### 2.3. Chip Operation

Biological Samples. The saccharomyces cerevisiae cells were dispersed in the cell medium with final cell densities of $4.5\times 10^{6}$, $8.2\times 10^{6}$, $1.18\times 10^{7}$, $1.57\times 10^{7}$, $1.96\times 10^{6}$, $2.35\times 10^{7}$, $2.74\times 10^{7}$, and $3.13\times 10^{7}$ cells/mL; each concentration was prepared with 2 mL, enough for producing all the droplets required for one test. The oil phase was prepared using Tetradecane (CAS 629-59-4, Aladdin, Shanghai, China) with 2% wt./wt. EM90 (CAS 144243-53-8, Degussa AG, Frankfurt, Germany) and 0.08% wt./wt. Triton-X 100 (CAS 9002-93-1, Biofroxx, Einhausen, Germany) as stabilizing surfactants.

The Formation of Droplets. Cell medium and oil solutions were loaded into Teflon tubing (Pureshi Chemistry, ID = 1 mm, OD = 1.6 mm) and then injected into the microfluidic device by AF1/AF1 Dual pressure pumps (Elveflow, Paris, France). MSF3 flow sensors (Elveflow, Paris, France) were used to monitor the flow rates of the two phases. The bright-field images were observed using an inverted microscope (Nikon Eclipse TI-S, Tokyo, Japan) equipped with a CCD camera (Nikon DS-QI 2). The recorded images were further processed with the help of NIS-Elements (Nikon, Tokyo, Japan). Before being pumped into the microfluidic channel, the cell suspension in its container was placed in an ultrasonic machine (FUYANG F-060SD, Shenzhen, China) for 2 min ultrasonic treatment. This step was to ensure that the cells were uniformly dispersed in the medium; at the same time, it could be observed that there was no cell adherence on the container’s bottom and walls. 

## 3. Results and Discussion

### 3.1. Development of the CFSE Step Microstructure

The unique feature of the proposed microfluidic CFSE device is its specific step microstructure with a sharp change in channel space, which is usually fabricated by multilayer lithography. Multilayer lithography is equivalent to multiple repetitions of single-layer lithography; it involves multiple photoresist coating, mask alignment, repeated UV exposure, and photoresist development, which is complicated, expensive, and time-consuming. In this study, to obtain a step microstructure, we used “punching” instead of “multilayer lithography”, verified as a straightforward and intelligent method.

As demonstrated in Figure 2a, the “punching fabrication” was carried out entirely manually without any auxiliary equipment such as a microscope. First, a bio-puncher was placed vertically on the PDMS piece to achieve a perfect alignment with the straight microchannel. Second, the bio-puncher was pushed down to penetrate the PDMS replica. The sharp edges can easily penetrate the PDMS replica to form a circular hole. Third, the PDMS debris was removed, thus developing a step microstructure and a cylindrical chamber (see Appendix A). 

This method has three benefits, as demonstrated below. First, the “punching” is easy to perform, avoiding the secondary photoresist coating, mask alignment, UV exposure, and photoresist development involved in multilayer lithography (see Appendix A). Second, “punching” is low-cost. It is finished only with the help of a bio-puncher, avoiding the high material and labor costs of “multilayer lithography”. Third, the “punching” is easy to perform. We gave training to three undergraduates (taking less than half an hour, including interpretations and demonstrations), after which they proved to be competent to fabricate such a CFSE device.

Since the punching fabrication is conducted manually, there may be alignment deviation between the punched hole and the straight microchannel. It is crucial to investigate the effects of punching positions on droplet production. Figure 2b demonstrates the three typical cases of the step microstructures from the “cutting” operation. The cylindrical holes are below (I), on (II), and above (III) the central line of the shallow straight microchannel, respectively. Figure 2c shows the corresponding optical images; it is worth noting that all holes could cut the shallow channels to create step microstructures even if offset or tilted. For 30 independent operations, the numbers of conditions I, II, and III were 5, 22, and 3, respectively. These fabricated chips were all successfully used to produce emulsion droplets, indicating that “punching fabrication” is a reliable method for developing a microfluidic CFSE device.

### 3.2. Droplet Generation

In the proposed device, we employ a microfluidic “co-flow step emulsification” (CFSE) approach to achieve droplet generation. Figure 3a shows the schematic diagram of the proposed 3D cell culture platform, consisting of a micropump, pressure controller, fluid tubes, CFSE chip, and inverted microscope. With the help of a micropump and a pressure controller, cell medium and oil were driven into the CFSE chip; they then formed a “co-flow” at reasonable flow rates. Under the continuous drive of pressure gas, the co-flow entered the deep cylindrical chamber from the shallow straight microchannel; it broke up into discrete droplets due to the Rayleigh–Plateau instability when crossing the step microstructure (see Appendix A). Figure 3b shows an optical photograph of the CFSE device and the resulting droplets. Here, the fluid flow rates were set at 1.28 μL/min (inner phase) and 0.49 μL/min (outer phase), respectively; the resulting droplets have a mean diameter of 80.38 μm and a standard deviation of 2.18 μm. As the oil phase is mainly used to separate the resulting droplets rather than shear the inner phase, the oil flow rate can maintain a relatively low value, resulting in an ultra-high droplet volume fraction (72%). Figure 3c shows the perfect encapsulation of cells in the droplet arrays. It was observed that the cells aggregated in the centre of the droplets in 14.91 ± 0.60 s; all cells in droplets could be photographed clearly. 

In addition, we have also investigated the thermal stability of the microfluidic CFSE device. Five independent droplet generation experiments were carried out using cell suspensions at 20, 25, 30, 35, and 40 °C; the resulting droplets had sizes of 80.78 ± 2.54 μm. Additionally, five other independent droplet generation experiments were carried out under different ambient temperatures (20, 25, 30, 35, and 40 °C); the resulting droplets had sizes of 82.16 ± 2.72 μm. Since the temperatures of most cell culture scenarios are in the range of 20~40 °C, the experiments above are sufficient to demonstrate the excellent thermal stability of the CFSE device.

Furthermore, we also investigated the chemical stability of the microfluidic CFSE device. For most cell culture experiments, cell suspensions are aqueous and they would not react with PDMS; thus, the CFSE device can be reused many times (at least 20 times for the saccharomyces cerevisiae cell culture experiments). Therefore, the CFSE device has excellent chemical stability to be applied in in droplet 3D cell culture (see Appendix A). 

### 3.3. Close-Packed Droplet Arrays 

The most important feature of this device is the ultra-high droplet volume fraction, which results in close-packed droplet arrays. As shown in Figure 3d, after CFSE droplet generation, the emulsion droplets moved along the circular edges and stacked together. With continuous perfusion, a steady droplet stream bulked the entire reservoir and the droplets arranged themselves layer by layer in a hexagonal pattern. The fabricated cylindrical pool, 6 mm in height and 4 mm in diameter, can accommodate approximately 150,000 droplets (*Φ* = 80.38 ± 2.18 μm, Q_inner-phase_ = 1.28 μL/min, Q_outer-phase_ = 0.49 μL/min). Each droplet can serve as an individual chamber for 3D cell culture, making large-scale and parallel cell culture assays possible.

Figure 3e shows the droplet arrays of six layers arranged from top to bottom. It can easily be seen that, in the horizontal plane, the droplet arrays were highly ordered in a hexagonal pattern. In the vertical plane, droplet arrays were staggered and achieved the most compact arrangement under the minimum energy principle [32]. During the experiments, all droplets remained stable and no droplet rupture or coalescence was observed in the cylindrical chamber. This can be attributed to the employment of surfactant EM90 and Triton-X 100, which can stabilize droplet emulsions for fairly long durations.

### 3.4. Three-Dimensional Cell Culture 

We have investigated the characteristics of the proposed system, including droplet generation stability, individual droplet volume, and cell-dispersing capability. An inverted microscope was used to photograph the droplets and the cells inside.

Droplet generation stability. We first studied the time-dependence of droplet quantities. Cell medium and oil flow rates were set at 1.28 μL/min and 0.49 μL/min. As shown in Figure 4a, approximately 12,900 droplets (*Φ* = 80.38 ± 2.18 μm) were produced in 160 s. The linear dependence indicates the stable frequency by which 81 droplets were generated per second. Thanks to the unlimited droplet production from continuous perfusion, the proposed cell culture platform is scalable. Second, we studied the monodispersity of droplets, and the corresponding data are shown in Figure 4b. From the plot of the measured droplet diameter versus running time from 0 to 8 minutes, the droplet sizes remained at 80.38 ± 2.18 μm, with excellent monodispersity. In addition, we have compared droplet generation in multiple devices; the 4.02% CV value indicates the excellent reproducibility of droplet generation in different CFSE devices.

Individual droplet volume. Droplet diameter can be regulated by fluid flow rates, making the culture system extraordinarily flexible and diverse. Figure 4c demonstrates the device’s capability to make droplets with custom sizes. The cell medium flow rate varied from 0.40 μL/min to 1.80 μL/min (the oil flow rate was kept at 0.70 μL/min). It is worth noting that droplet size is well-regulated by the flow rates of the two phases; the insert graphs also support the same conclusion—that droplet diameter expanded as the inner phase flow rate increased. Adjustable droplet volume is of great significance; for instance, small-sized droplets with a high surface-to-volume ratio can improve nutrient exchange efficiency with the outer environment, which could be useful in cell culture experiments with faster breeding cycles. Large-sized droplets with a low surface-to-volume ratio can effectively reduce droplet evaporation, making them suitable for longer encapsulation and culture (see Appendix A).

Cell-dispersing capability. To verify the cell-dispersing capability of the proposed device, we used cell mediums of different concentrations ($4.5\times 10^{6}$, $8.2\times 10^{6}$, $1.18\times 10^{7}$, $1.57\times 10^{7}$, $1.96\times 10^{6}$, $2.35\times 10^{7}$, $2.74\times 10^{7}$, and $3.13\times 10^{7}$ cells/mL). For each dispersing experiment, it was observed that within 3 min approximately 13,000 droplets (81.51 ± 2.32 μm in diameter, CV < 5%, Q_inner-phase_ = 1.31 μL/min, Q_outer-phase_ = 0.50 μL/min) were generated in the chip. The number of cells per droplet was counted, and the corresponding data are shown in Figure 4d. The linear dependency indicates that the cell number per droplet is highly dependent on the cell concentrations in the medium. The same conclusion can also be demonstrated in the four insert graphs: 1, 3, 5, and 7 cells were encapsulated in an individual droplet as the cell density increased from $4.5\times 10^{6}$ to $2.74\times 10^{7}$ cells/mL. Therefore, the proposed device has excellent cell-dispersing capability.

Finally, we applied the proposed system in the 3D culture of saccharomyces cerevisiae cells. With fixed conditions, namely, 65.4 pL droplet volume (*Φ* = 50.15 ± 1.13 μm), $1.61\times 10^{7}$ cells/mL cell density, and a 200 minutes incubation time, saccharomyces cerevisiae cells were cultured in close-packed droplet arrays with an occurrence rate for the proliferation of 80.34 ± 3.77%, almost the same as the commercial culture platform (see Appendix A). Figure 4e shows the long-term observations of the budding reproduction of a saccharomyces cerevisiae cell. It is worth noting that no droplet rupture or fusion was observed during the culture process.

## 4. Conclusions

In summary, we have proposed a microfluidic CFSE device for achieving large-scale and high-throughput in droplet cell culture. Three modules, including droplet generation, droplet self-assembly, and in situ cell culture, are integrated in a single device. With the microfluidic punching fabrication method, an elaborate step microstructure was developed to achieve “co-flow step emulsification” for the production of close-packed droplet arrays. The resulting droplet arrays not only achieved the perfect encapsulation of cells but also had an ultra-high-volume fraction (72%) that can be used in large-scale and high-throughput cell culture. To evaluate the 3D cell culture performance of our platform, saccharomyces cerevisiae cells were employed and encapsulated in droplet arrays with individual volumes of 65.4 pL (*Φ* = 50.15 ± 1.13 μm) for 200 min. With an occurrence rate for the proliferation of 80.34 ± 3.77%, the cell culture has been proved successful, and the proposed device can meet the requirements of 3D cell culture. Moreover, the cost of each microfluidic CFSE chip is about USD 3, making it a low-cost approach for 3D cell culture.

Our CFSE device does have some shortcomings, however. First, the punching fabrication for developing a step microstructure is still skill-dependent, and researchers need to take a short course of training to construct reliable devices. Second, the automation of the proposed system is inadequate, including reagent pumping, imaging, and integration of the instrumental device. This could, however, be solved in the future by engineering approaches. Third, the biological applications involved in this paper are relatively simple; nevertheless, the presented experiments have demonstrated the system characteristics and we are considering clinical trials by means of which to conduct further biological studies in the future.

In a nutshell, owing to the characteristics of a simple system setup, low consumption rates, an ultra-high droplet volume fraction, and mass droplet production, the present co-flow step emulsification device is expected to expand the range of cell culture applications, including tissue engineering and high-throughput drug screening.

## Figures and Tables

**Figure 1 biosensors-12-00350-f001:**
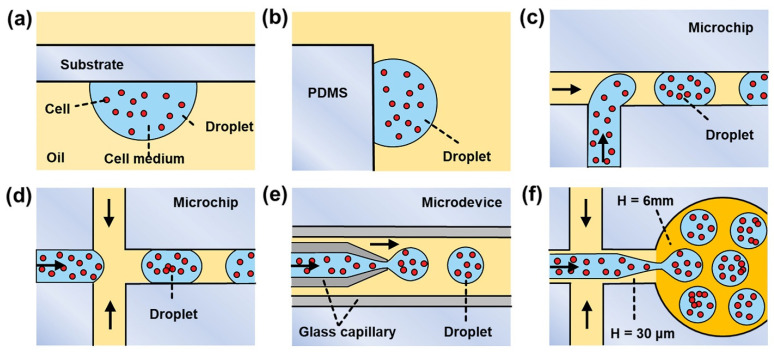
Typical microfluidic 3D cell culture methods: (**a**,**b**) hanging drop; (**c**) T-junction; (**d**) flow-focusing; (**e**) coxial flow; (**f**) co-flow step emulsification (CFSE).

**Figure 2 biosensors-12-00350-f002:**
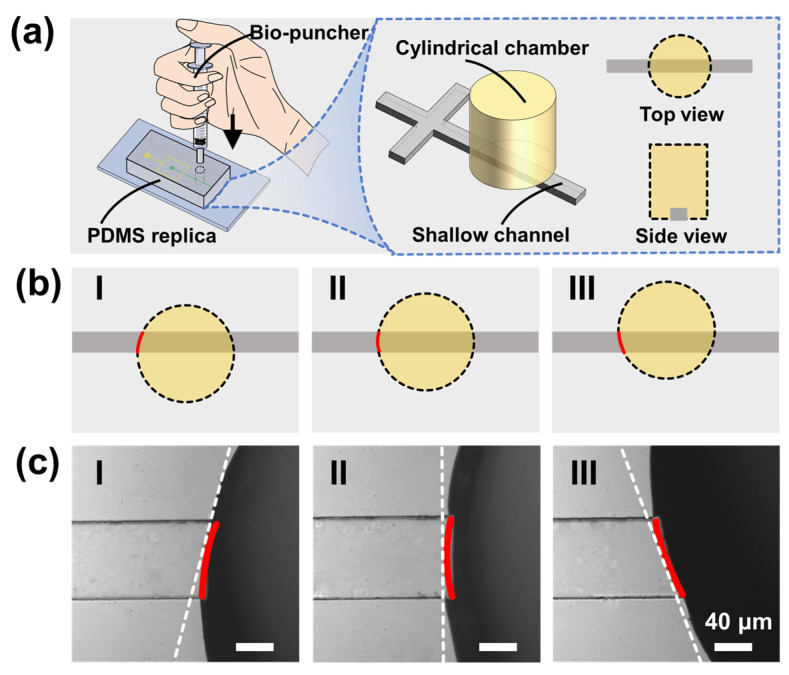
Punching fabrication. (**a**) I: Schematic diagram of the punching operation (**b**) Three typical illustrations of the step microstructures. I: The cylindrical hole is below the central line of the straight microchannel. II: The cylindrical hole is right on the central line of the straight microchannel. III: The cylindrical hole is above the central line of the straight microchannel. (**c**) Microscopic images of the step boundaries, corresponding to the illustrations above. It is worth noting that the bio-puncher can smoothly cut the straight channels without edge-tearing or cracking.

**Figure 3 biosensors-12-00350-f003:**
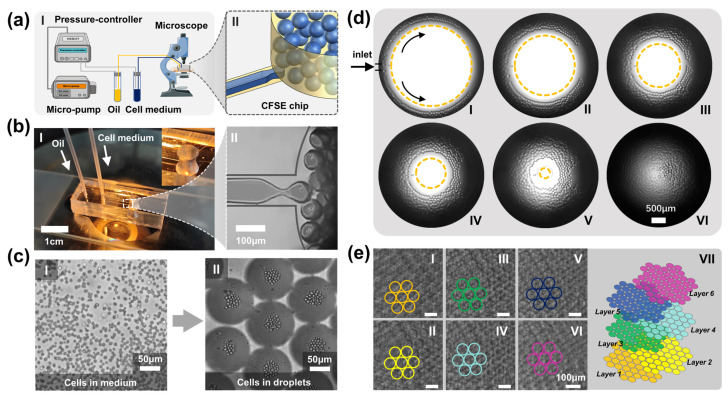
CFSE-based droplet generation. (**a**) Schematic of the 3D cell culture system. I: The system consists of a micropump, pressure controller, fluid tubes, CFSE chip, and inverted microscope; II: Schematic diagram of the “co-flow step emulsification”. (**b**) Optical photograph of the microfluidic CFSE device. I: Cell medium and oil were pumped into the microchannels; II: The cell medium broke into emulsion droplets. (**c**) Cell clusters in medium (I) and in droplets (II). (**d**) The resulting droplets bulk the cylindrical reservoir within several minutes with an ultra-high-volume fraction (φ = 72%, Q_inner-phase_ = 1.28 μL/min, Q_outer-phase_ = 0.49 μL/min, *Φ* = 80.38 ± 2.18 μm). I–VI shows the droplet perfusion process in the cylindrical storage reservoir in 30 seconds. (**e**) The close-packed droplet arrays arrange themselves in a hexagonal pattern. I–VI demonstrate the droplet arrays of six layers arranged from top to bottom. VII demonstrates the stacked status of the droplet arrays.

**Figure 4 biosensors-12-00350-f004:**
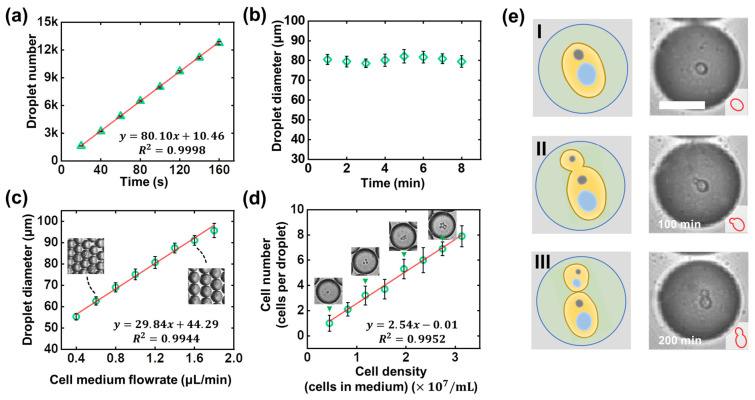
Investigation of the CFSE-based 3D cell culture system. (**a**) Time-dependence of droplet numbers. (**b**) Plot of emulsion droplet diameter as a function of running time, to show the time-stability of the microfluidic CFSE device. The cell medium and oil flow rates were set at 1.28 μL/min and 0.49 μL/min. (**c**) Measured diameters of the droplets were plotted against the different cell medium flowrate conditions. The inset microscopic images show the droplets with diameters of 62.62 ± 1.95 μm and 91.09 ± 2.26 μm, respectively. (**d**) The relationship between cell number per droplet and cell density in the medium. (**e**) Illustration and microscopic images of the proliferation process of saccharomyces cerevisiae cell. I: A saccharomyces cerevisiae cell was encapsulated in a droplet; II: The saccharomyces cerevisiae cell began to bud and proliferate; III: The saccharomyces cerevisiae cell kept proliferating for 200 min. Scale bar: 30 μm.

## Data Availability

The data that support the findings of this study are available from the corresponding author upon reasonable request.

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
