# Peer review of "Easy-to-Operate Co-Flow Step Emulsification Device for High-Throughput Three-Dimensional Cell Culture"

_biosensors, 2022, doi:10.3390/bios12050350_

Round 1

Reviewer 1 Report

The following comments can help the authors to improve the manuscript before an acceptance:

1. A more comprehensive explanation of the working mechanism of the device is needed.
2. The authors mentioned about low cost, but I do not see the estimated price of the process and the device (also compared to the costs when using other methods)
3. How was the droplet's size measured?
4. How long can the cell culture take place with this method to achieve confluency?
5. Figure 3 A: since you have a cell suspension in a container before pumping into the microfluidic channel, there are a couple of questions:
 a. Did the cell adhere to the container's bottom and wall?
 b. What were the flow rates was applied to pump the medium and cells?
 c. How to obtain or reuse the cells after culturing?

Reviewer 2 Report

The authors presented a novel microfluidic device for 3D cell culture based on “co-flow step emulsification (CFSE). They reported that the proposed microfluidic device is economical and easy to operate owing to the succinct “punching” chip fabrication method. Furthermore, the proposed cell culture method is flexible, a benefit of the regulable droplet diameters and the scalable droplet numbers. They successfully applied this system to saccharomyces cerevisiae cell culture. The manuscript is well organized. The presented paper is interesting, but the following corrections should be done before publishing.

Below are my concerns and suggestions to improve the manuscript,

The abstract needs to be highly quantitative.

The introduction section is missing a piece of brief information on the saccharomyces cerevisiae cell.

The conclusion needs to be highly quantitative and should be discussed in more detail.

Please provide more details about section 3.2 Droplet generation.

Please provide more details about the detection time, and the sample volume. The authors must explain the advantages of the developed technique. 

Reproducibility studies are very important for this device. The authors must explain the advantages. 

Please, explain the thermal stability and chemical stability of the microfluidic CFSE device.

Round 2

Reviewer 1 Report

I do not have further comments.

Reviewer 2 Report

Authors have carefully checked and modified this manuscript. 
Therefore, the article can be accepted in this form in the journal "Biosensors".